# The Relationship between Lower Extremity Functional Performance and Balance after Anterior Cruciate Ligament Reconstruction: Results of Patients Treated with the Modified All-Inside Technique

**DOI:** 10.3390/jpm13030466

**Published:** 2023-03-02

**Authors:** Nizamettin Güzel, Ahmet Serhat Genç, Ali Kerim Yılmaz, Lokman Kehribar

**Affiliations:** 1Department of Orthopedics and Traumatology, Samsun Training and Research Hospital, 55090 Samsun, Türkiye; 2Departments of Recreation, Faculty of Yaşar Doğu Sport Sciences, Ondokuz Mayıs University, 55100 Samsun, Türkiye; 3Department of Orthopedics and Traumatology, Samsun University, 55090 Samsun, Türkiye

**Keywords:** anterior cruciate ligament reconstruction, modified all inside, hop test, balance, return to sport

## Abstract

Background and Objectives: Anterior cruciate ligament (ACL) ruptures are common injuries, and ACL reconstruction (ACLR) is among the most common surgical procedures in sports surgery. Our research aims to compare the 6-month post-operative results of the modified all-inside (MAI) ACLR technique, single leg hop tests (SLHT), and Y balance tests applied in different directions on the operated and non-operated sides. Materials and Methods: A retrospective cohort of 22 male recreational athletes who underwent MAI ACLR techniques performed by the same surgeon were evaluated. The functional knee strengths of the participants on the operated and non-operated sides were evaluated with five different tests of SLHTs: single hop for distance (SH), triple hop for distance (TH), crossover triple hop for distance (CH), medial side triple hop for distance (MSTH), and medial rotation (90°) with hop for distance (MRH). Their dynamic balance was evaluated with the Y balance Test. Results: Compared to pre-operative levels, there was a significant improvement in the mean Lysholm, Tegner, and IKDC scores during the post-operative period (*p* < 0.05). There was a difference between SH, THD, CHD, MSTH, and MRH on the operated and non-operative sides (*p* < 0.05). There was no difference between Y balance scores on the operated and non-operative sides, and there were no differences between LSI scores resulting from SLHTs (*p* > 0.05). There were no significant relationships between YBT (composite scores) and SH, TH, CH, MSTH, and MRH distances in the healthy leg (*p* > 0.05), but a significant correlation with only CH in the ACL leg (*p* < 0.05). Conclusions: Our research shows that sixth-month post-operative SLHT findings were lower on the ACL side compared to the healthy side in patients tested with the MAI ACLR technique. However, when these scores are evaluated in terms of balance, it can be seen that both sides reveal similar findings. The similarity of LSIs in SLHTs applied in different directions, and balance scores of ACL and healthy sides revealed that the MAI technique is also an ACLR technique that can be used in athletes from a functional point of view.

## 1. Introduction

The anterior cruciate ligament (ACL) is one of the most common orthopedic injury sites, and the mechanisms of injury have been associated with rotational movements, including sudden acceleration and deceleration, sudden changes in direction when the foot is in a fixed position, and external blows to the knee during sports [1,2]. ACL reconstruction (ACLR) involves the preparation of grafts and their restructuring through tunnels opened into the femur and tibia to restore ligament function in individuals with anterior cruciate ligament insufficiency and reduce the risk of future osteoarthritis and future degeneration of other soft tissues of the knee joint [1,3]. The most commonly used ACLR graft types today are the quadriceps tendon (QT), patellar tendon (PTG), and semitendinosus/gracilis (ST/G) tendons [4]. In addition, the so-called “all-inside” technique, with only ST and fourfold (4ST) grafting, has become widespread recently [5]. The all-inside technique has many advantages, but the modified all-inside (MAI) technique was developed by Mahirogullari et al. [6] due to the need to use a special drill while creating the socket, the problem of adjusting the socket depth, and the high cost of the all-inside technique [7,8]. In the MAI technique, the ST graft, prepared by folding the ST tendon four times, is fixed to the tibia and femur with a suspended suspension [9].

After ACLR, patients experience problems such as pain, impaired knee function, and especially quadriceps muscle weakness and atrophy [10,11]. It is known that there are proprioceptive receptors in the ACL, and these receptors form the reflex protective arc for stable muscle contractions [12,13]. When the ACL is ruptured, sensory stimuli from these mechanoreceptors in the knee to the central nervous system are absent, leading to a loss in the stabilization ability of the lower extremity, and a significant loss in proprioception occurs secondary to the instability of the knee [14]. Researchers have reported a loss of proprioceptive functions after ACL injuries [15,16,17]. For this reason, criteria such as functional single-leg hop tests (SLHT) and balance tests are of great importance and are widely used in decision-making for rehabilitation processes and prior to returning to sports (RTS) after ACLR [18]. Post-ACLR balance is a critical factor for sports performance and the activities of daily living [19]. Y balance test (YBT) is one of the dynamic balance tests frequently used in clinical and research settings to evaluate lower extremity function [20,21]. It is frequently used to evaluate dynamic balance disorders that may occur with lower extremity injuries, such as ACL injury and patellofemoral pain syndrome [22,23,24,25]. Lower SCT scores were associated with an increased risk of ACL disability, ACL deficiency, and ACLR [25,26,27,28,29,30].

SLHTs are widely used to evaluate athletes’ functional status after ACLR, reveal limb asymmetries between the operated and non-operated sides, and follow the limb’s developments [31,32,33]. The most significant limitation of conventionally applied SLHTs is that they mainly consist of forward movements [31,34]. Researchers have stated that multi-directional tests are important in addition to SLHTs performed in the forward direction, especially when returning to sports after injury [31,35]. Studies have also reported increased asymmetry rates in multi-directional tests compared to conventionally applied SLHTs [9,36]. For this reason, it can be said that multi-directional hop tests and traditional SLHTs after ACLR play a critical role in determining rehabilitation and return to sports time [9]. A study conducted after ACLR reported that decreased quadriceps strength on the operated side was associated with lower distances in hop tests [37]. Researchers examined limb asymmetries after ACLR and emphasized the importance of considering this factor [38,39]. These results suggest that it may be important to consider SLHTs to decide on RTS after ACLR [40].

Based on all this information, our study evaluates the results of SLHT applied both in the forward direction and in the medial and rotational directions, the resulting limb asymmetries and SCT results in patients with a 6-month post-operative ACLR history who underwent the MAI ACLR technique and compared them with the healthy sides. It was hypothesized that similar SLHT and SCT results would be obtained between the ACLR and non-operated sides of the research subjects.

## 2. Materials and Methods

### 2.1. Experimental Approach and Patients 

This study used a retrospective institutional registry of patients treated for ACL tears at an academic medical center. We identified 22 active (recreational) male patients aged between 18 and 35 from a sequential case series from January 2020 to December 2021 with a diagnosis of isolated ACL tear who were treated with MAI (hamstring autograft) anterior cruciate ligament reconstruction (ACLR). Before diagnosing ACL rupture, patients presented with weakness, atrophy, pain, swelling, and slipping in the knee at the examination. Diagnosis of ACL tear was made primarily by the Lachman test and MR imaging. The number of subjects was determined by G*Power 3.1. According to the results, it would be sufficient to work with 19 patients (effect size r: 0.88, lower and upper critical p: 0.54, real power: 0.94). However, our study was completed with 22 subjects to examine the research findings. None of them was a professional sports patient, and participants consisted of those who participated in recreational activities only. Inclusion criteria were undergoing MAI (hamstring autograft) anterior cruciate ligament reconstruction (ACLR) and having no history of any other neuromuscular or musculoskeletal injury and contralateral knee surgery or injury. Exclusion criteria were non-compliance with the rehabilitation calendar, failure to complete patient-reported outcomes (PROs), SLHT, and YBTs, or exposure to various complications at the follow-up (Figure 1). The average time between the injury and the surgery was about two months. Detailed characteristics of the cohort are shown in Table 1. This study was approved by the local ethics committee with the number “SÜKAEK-2023-3/14”, and all patients gave signed informed consent.

PROs (Lysholm, Tegner, International Knee Documentation Committee (IKDC)) (pre- and post-operative), single-leg hop test (SLHT) distances, and YBT scores obtained 6–7 months after ACLR were taken from routine clinical procedures. All reported evaluations were routinely part of the ongoing clinical procedure. The MAI ACLR technique was performed by a single surgeon specializing in soft tissue knee injuries, and data were collected from a single center. Post-surgical treatment was provided with a mandatory rehabilitation protocol as a routine clinical procedure. The rehabilitation program was the same for all patients and was followed in a controlled manner. 

### 2.2. Surgical Treatment (Modified All-Inside Technique) 

All patients underwent anatomical single-bundle ACL reconstruction using adjustable suspension fixation with a quadruple semitendinosus tendon autograft. The retrieved semitendinosus tendon (24–28 cm long) was quadrupled with adjustable loop cortical suspension fixation (Lift Loop External; Orthomed) on the tibial end and fixed loop button system (Femobutton; Orthomed) on the femoral side. The Lift Loop Outer fastening system consists of a 20 mm wide titanium button and two loops controlled by a knotless locking mechanism. The femobutton comprises a 10 mm wide titanium button and a continuous loop available in 6 lengths (15–40 mm). The anatomical femoral tunnel was carved from the anteromedial portal. First, a full bone tunnel was drilled over the guide pin using a 4.5 mm drill. The tunnel length was then measured, and a socket was opened using a router the same size as the graft, considering the 6 to 8 mm EndoButton ‘flip’ movement distance. A complete outside-in tibial tunnel was created at the anatomical footprint’s central location. The graft was passed from the intra-articular space to the tibial and femoral tunnels, and the graft knee was stretched at 20° of flexion and flexed and extended 30 times. Graft tightness was then examined with a probe. Finally, the entire structure was re-tensioned on the tibial side, and an additional ligation was made on the adjustable suspension fixation device using a non-slip knot [6]. Figure 2 shows intraoperative fixation. Figure 3 presents a post-operative roentgenography showing the use of the materials. 

### 2.3. Procedures 

All patients refrained from performing the high-intensity exercise the day before measurements to avoid the effects of cumulative muscle fatigue. The patients completed the YBT and SLHT functional tests consisting of anterior (ANT), posterolateral (PL), and posteromedial (PM) directions for balance and the single hop for distance (SH), triple hop for distance (TH), crossover triple hop for distance (CH), medial side triple hop for distance (MSTH), and medial rotation (90°) hop for distance (MRH) assessments. Trials were performed prior to measurements to familiarize patients with the tests fully. There were 10-min intervals between the YBT and SLHT, and all patients performed a 5-min warm-up before the tests. Figure 4 shows the setup of both tests.

#### 2.3.1. Anthropometric Measures 

Anthropometric information, such as height, weight, body mass index (BMI), thigh length, and lower limb length was collected. Thigh length was measured from the anterior superior iliac spine to the superolateral border of the patella. The lower limb length was measured bilaterally in supine position from the anterior superior iliac spine to the most distal aspect of the medial malleolus to the nearest half-centimeter [41].

#### 2.3.2. Y Balance Test (YBT) 

The YBT measures dynamic balance in the ANT, PL, and PM directions. The ground was marked in 3 different directions (⅄-shape) with a 15 cm wide tape; the angles between the anterior stripe and both posterior (posteromedial and posterolateral) stripes were 135 degrees, with 90 degrees between the two posterior stripes. Participants were instructed to place the stance foot in the marked position (zero-mark position) of the anterior, reach as far as possible in the reaching direction determined with the standing foot at the starting line, and then return the reaching foot to the starting position. ANT reach distance was measured from the toe of the stance foot in the starting position to the point reached, and PL and PM were measured from the heel of the stance foot to the point reached. The supervisor visually checked positions. Trials were canceled and repeated if the patient: 1) could not maintain his balance, and 2) failed to return the reach leg to the side of the stance leg after achieving maximal reach distance. Patients performed three trials for each direction (ANT, PL, and PM) on both legs with bare feet, and the best of the trials was recorded. Reach distances were normalized to anatomical leg length and expressed as a percentage (reach distance/limb length × 100). Composite normalized reach distance was computed for each leg as (ANT + PM + PL)/(3 × limb length) × 100 [20,27].

#### 2.3.3. Single Leg Hop Tests (SLHT) 

The SLHT included SH, TH, CH, MSTH, and MRH over a single line with maximum effort [42,43]. Kivlan et al. have demonstrated that SLHTs have good test–retest reliability in patients after ACL reconstruction [43]. 

The ground was marked with a 6 m long, 15 cm wide tape running perpendicular to the start and finish lines. Before starting the tests, an examiner explained and demonstrated the hop test procedures. Patients performed one practice trial (familiarizing practice) per limb, followed by three trials measured and recorded. Both limbs were tested, and subjects were not restricted in their arm movements. A 30-s rest interval was used after each attempt of the same hop tests. To minimize fatigue, a rest period of up to two minutes was given between different types of hop tests. 

Patients began each trial behind a marked starting line. The best-recorded performance was the distance between the starting line and the heel of the foot where it landed at the end of the task. All hop tests were considered successful if the landings were stable. The post-jump landing was approved when it was under the participant’s full control and on the tested limb. The test was repeated if the participant lost balance, touched the wall, or had additional bounces after landing [9]. The examiner visually checked this position.

### 2.4. Statistical Analysis 

All statistical analyses were conducted using SPSS for Windows version 21 (IBM Inc., Chicago, Illinois). Descriptive data were presented as the mean, standard deviation (SD), median (Med), minimum (Min), or maximum (Max). The data were checked for normality using the Shapiro–Wilk test and were examined for kurtosis and skewness. Differences between PROs (Lysholm, IKDC, Tegner), YBT (ANT, PL, PM), and SLHT (SL, TH, CH, MSTH, MRH) distances were compared with paired samples *t*-test or Wilcoxon test. The ANOVA was also used to evaluate the limb symmetry index (LSI) differences. The Spearman’s rank correlation test was used to calculate correlations between YBT (Composite score) and SLHT scores. Effect sizes were determined using the Cohen d, which defines 0.10, 0.30, and 0.50 as small, moderate, and large, respectively [44]. Significance was set at *p* < 0.05, with associated 95% confidence intervals. 

## 3. Results

Table 1 summarizes the patient characteristics and clinical variables. There was no significant difference between pre-and post-operative in the thigh length and lower limb-length (*p* > 0.05), and pre- and post-operative Lysholm, IKDC, and Tegner scores are presented in Table 2. All PROs significantly improved after the operation (Lysholm, *p* < 0.001; IKDC, *p* < 0.001; Tegner, *p* = 0.002).

Table 3 compares anterior, posterolateral, posteromedial, and composite scores for 232 the operated and healthy legs. Although the all-direction reach scores of the operated leg 233 were lower than the healthy leg, it was no statistically significant differences (*p* > 0.05).

Significant differences were found in all tests considering the SLHT distances for the operated and healthy leg. The healthy leg had slightly higher scores compared to the operated leg in SH (114.14 ± 25.14 vs. 103.36 ± 29.21; *p* = 0.006), TH (392.05 ± 87.92 vs. 362.73 ± 88.78; *p* = 0.007), CH (340.59 ± 89.12 vs. 319.14 ± 75.53; *p* = 0.038), MSTH (310.27 ± 74.79 vs. 293.59 ± 70.16; *p* = 0.049), and MRH (106.41 ± 30.54 vs. 97.05 ± 34.04; *p* = 0.030), respectively (Figure 5).

No statistically significant differences were found between the LSI scores (*p* > 0.05) (Table 4).

There were no significant relationships between YBT (composite scores) and SH, TH, CH, MSTH, and MRH distances in the healthy leg (*p* > 0.05), but significant correlation with only CH in the ACL leg (*p* < 0.013; r =−0.520) (Figure 6).

## 4. Discussion

When the major findings of our study were evaluated, the results were as follows; The MAI ACL reconstruction technique revealed an improvement between the pre-operative and post-operative findings in terms of Tegner, Lysholm, and IKDC scores, and the sixth-month post-operative findings of the patients’ SH, TH, CH, MSTH, and MRH performances were lower in the ACLR sides compared to the healthy sides. When evaluated in terms of LSIs, it was determined that the resulting rates were within the normal norm ranges. Finally, when the patients were evaluated in balance scores, sixth-month post-operative findings were similar between ACLR and healthy knee.

Due to the damage to the proprioceptive receptors after ACL tears, the stimuli from the mechanoreceptors to the nervous system disappear, which may lead to a loss of balance [12,13,14,15,16,17]. In addition, the loss of muscle strength and atrophy [10,11] in the quadriceps and hamstring muscles after ACLR, especially after using hamstring autografts, led the researchers to evaluate these two critical parameters (lower extremity strength and balance). In a study evaluating balance and lower extremity function after ACLR with different graft types (PT, hamstring autograft (HT), and allograft (AG)), it was reported that the ACLR group had a decrease in anterior reach distance in balance in both the healthy and operated sides compared to the healthy group. It was reported that there was no significance in terms of balance in the operated and non-operated sides, and similar LSI rates were reported in normal norm ranges in SLHTs (SH, TH, CHD). When evaluated regarding graft types, it was reported that only the PT graft showed a decrease in PL reach distances compared to HT [45]. In their study, Bulow et al. [46] did not report any difference in ANT, PL, and PM parameters in the Y balance test on a healthy group diagnosed with an ACL tear but not undergoing reconstruction. In electromyography (EMG) studies, they stated that while reaching the ANT and PM directions during the Y balance test, the vastus medialis muscle exhibited voluntary contractions ranging from 66% to more than 100%, and in this case, the knee extensor strength is of great importance for movements that require dynamic balance [47,48]. In the literature, studies conducted on healthy subjects with an ACL injury and ACLR and healthy subjects used less reliable dynamic balance test protocols than clinical tests such as the Y balance test [23,25,49]. Researchers found different findings for ANT access in these studies between healthy controls, participants with an ACL injury, and participants with ACLR. The researchers reported the findings as follows: the performances and kinematic profiles of the subjects at ANT, PL, and PM distances were different and each test loaded the participant with different neuromuscular demands and postural control strategies [50,51,52]. When the results of our current research and similar studies were evaluated together, it was revealed that the graft types used in ACLR, the physical activity status of the subjects, and the anthropometric characteristics of the subjects were functional. However, different forms should be used in dynamic balance tests, such as Y-balance, which has a high margin of error. To eliminate inconsistencies in the results, the results obtained together with clinical and neuromuscular tests may be more valuable. This is especially important for a clear understanding of the results in patients with ACL tears or who have undergone ACLR in different graft types. Our current study, one of the first studies in which balance was evaluated using the MAI technique, did not reveal any difference between the operated and non-operated sides in three different directions (ANT, PL, and PM) regarding balance. Our current research revealed that the CS of the Y balance test was less than 80% of the participants, which is to be expected from recreational athletes.

Functionally, SLHTs used to determine the knee functions and RTS duration of the subjects after ACLR are tests that can be easily used to reveal the differences between the operated and non-operated sides. Researchers have reported that LSI rates calculated from SLHTs are 10%–15% in healthy limbs [53]. When the studies on patient groups who had ACLR were examined, a study was found in which SLHTs were evaluated in groups who underwent ACLR with the MAI technique. In this study, researchers found statistical significance only in the SH test in five different SLHT (SH, TH, CH, MRH, and MSTH) tests and reported that LSI rates were within normal norm ranges in all tests [9]. A study conducted with the conventional HT ACLR technique measured the SH of the subjects at the 6th and 12th months and found LSI rates of >85 and above in most of the group [54]. Another study in elite football players evaluated SH and CH strengths after post-operative ACLR and reported that LSI ratios did not reach >90, and the difference between the two limbs was not within the re-injury risk range [55]. As in balance, it is known that the ACLR side reveals very low strength rates compared to the intact side, especially between extensor forces and tests that give clinical results, such as isokinetic and functional tests such as SLHTs in knee strength [55]. The findings in our current research and in other studies suggest that the ACLR technique and rehabilitation processes applied especially for retrospective design studies result from their contribution to the healing processes. As a matter of fact, the recovery processes of sedentary and recreational athletes may vary compared to elite athletes, and the findings and literature findings support this situation. However, most researchers report that at least two SLHTs applied in different directions can reveal significant results for RTS [8,56]. When evaluated with these results, the 6-month post-operative findings obtained from five different SLHTs measured in patients in whom the MAI ACLR technique was applied in our study revealed significant differences in distances. However, the fact that the LSI ratios were in the range of <10% proves that the MAI technique is viable in terms of functional lower extremity performance. The researchers stated that although statistical differences were not observed in LSIs, there may be differences in clinical decision-making depending on the threshold values. As a matter of fact, in the study conducted, while a similarity between the limbs was found at the level of >90% in all of the subjects in traditional SLHTs, >90% similarity was found in only 68.8% of the subjects in the medial and rotational hop tests [8]. Similar findings were also emphasized in studies and review studies on healthy subjects [57,58]. Our current study observed the lowest LSI rates results, especially in the MRH test. All these results clearly show that using multi-directional SLHTs is of great importance in decision-making for post-ACLR RTS. Although the reason why multi-directional SLHTs cause high asymmetry is still debated, the researchers emphasized that limb biomechanics may vary according to the direction of jump and descent as well as dynamic postural stability, and that hip abduction and medial rotation and knee valgus movement may be restricted during descent after the multi-directional jump. [43,59,60,61,62].

Regarding ACL surgery and graft types, the “all-inside” technique has many advantages, such as using a single tendon and creating a socket instead of a tunnel on the tibial side. The disadvantages are evaluated as the need to use special burs on the tibial side, the creation of the socket, the limited margin of error in the adjustment of the socket depth, the need to place the graft from the portal and the higher cost. Considering the abovementioned disadvantages, the “MAI” technique was developed for ACLR. This technique has the advantages of the all-inside technique but does not have the disadvantages mentioned above.

Our research is the first study to evaluate the MAI technique’s post-operative SLHT and balance results. In this respect, it is important to evaluate knee proprioception, stabilization, and strength together after an ACL injury. However, our study has a limitation in that it does not include tests involving fast turns and running, where ACL injuries often occur. In addition, the lack of other graft types and the absence of a control group are the main limitations of our study. In addition, one of the important limitations of our study is that the subjects did not feel any fatigue in the study. ACL injuries may occur due to voluntary or involuntary sudden movements at certain fatigue levels. It is important to plan studies by considering these limitations in future studies.

## 5. Conclusions

As a result, balance scores of patients who underwent MAI ACLR technique were similar on the operated and non-operated sides. In addition, although there were differences between both parties in SLHTs, this rate was not negatively reflected in LSIs. The fact that the CS scores were below 80% in the Y balance test and that the LSI scores were within the normal norm ranges reveals the outcome of the surgical technique and rehabilitation program applied, and that the non-operated sides were the least capable of the operated sides. Moreover, when supported by the literature, these findings are expected results for recreational athletes. In the future, prospective design studies on athletes, sedentary individuals, and recreational athletes, and the MAI technique’s short-, medium-, and long-term findings will be evaluated. This will contribute to the literature investigating whether this technique is viable in subject groups. At the same time, clinical and biomechanical tests and functional tests will provide clear information about whether this technique harms the movement structure in patients to whom the MAI technique is applied.

## Figures and Tables

**Figure 1 jpm-13-00466-f001:**
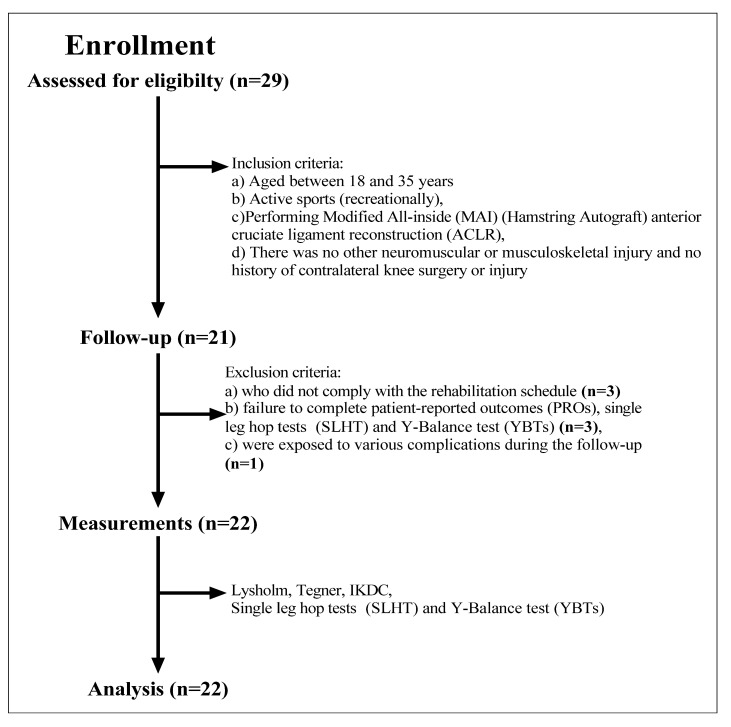
Flow chart diagram: description of the study population and inclusion and exclusion criteria.

**Figure 2 jpm-13-00466-f002:**
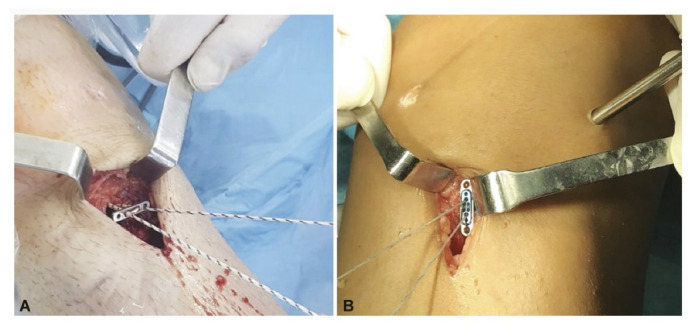
The graft was fixed to the tibial side with a suspended large button. (**A**) The graft was fixed to the tibial side with a Lift Loop. (**B**) The graft was fixed to the tibial side with an Ultrabutton plus Xtendobutton combination.

**Figure 3 jpm-13-00466-f003:**
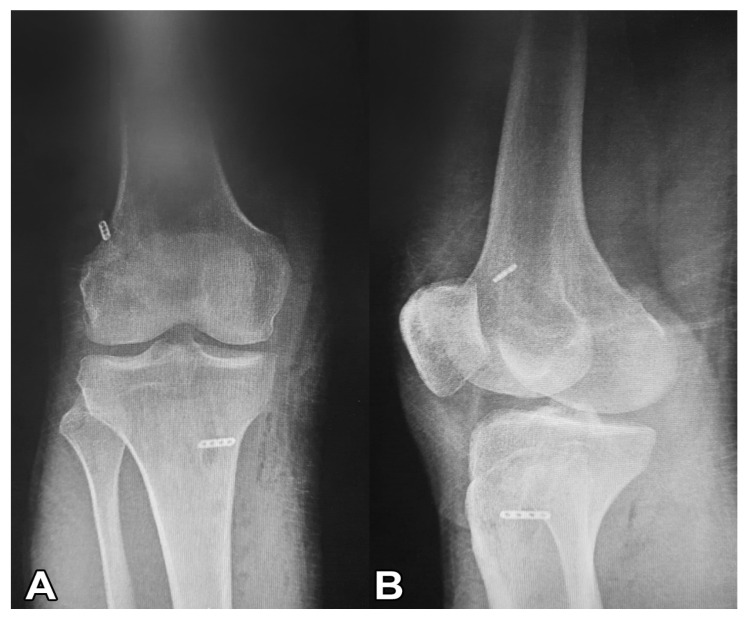
Post-operative roentgenography showing the use of the materials. (**A**) Post-operative roentgenography showing the use of Femobutton plus Lift Loop external. (**B**) Post-operative roentgenography shows three materials, Endobutton CL, Ultrabutton þ, and Xtendobutton in combination.

**Figure 4 jpm-13-00466-f004:**
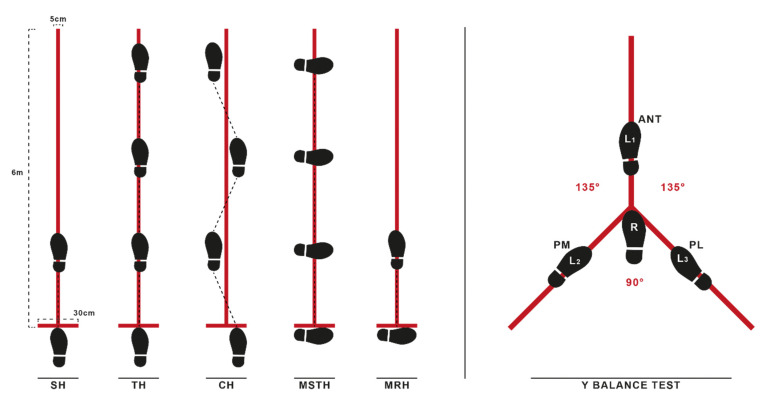
Demonstrations of SLHTs and Y balance tests. SH single leg hop for distance; TH triple hop for distance; CH crossover hop for distance; MSTH medial side triple hop for distance; MRH medial rotation (90°) hop for distance; ANT anterior; PL posterolateral; PM posteromedial.

**Figure 5 jpm-13-00466-f005:**
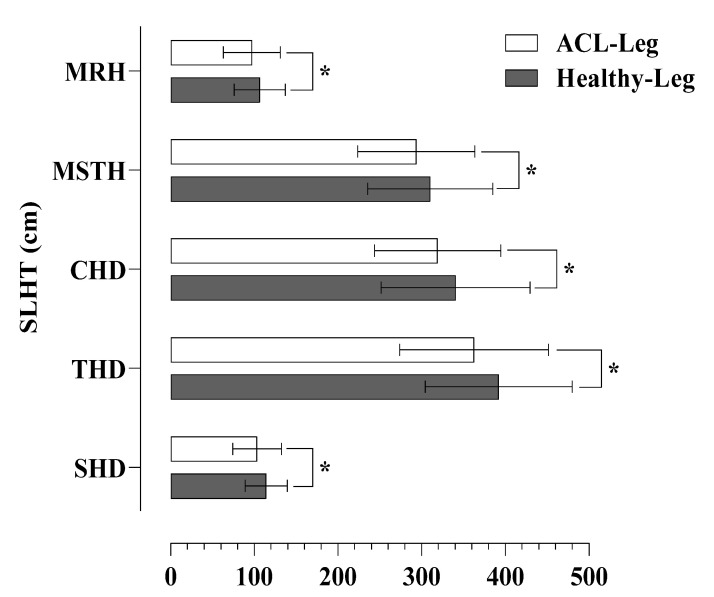
Differences in SLTH distances between operated (ACL-Leg) and healthy leg. * *p* < 0.05.

**Figure 6 jpm-13-00466-f006:**
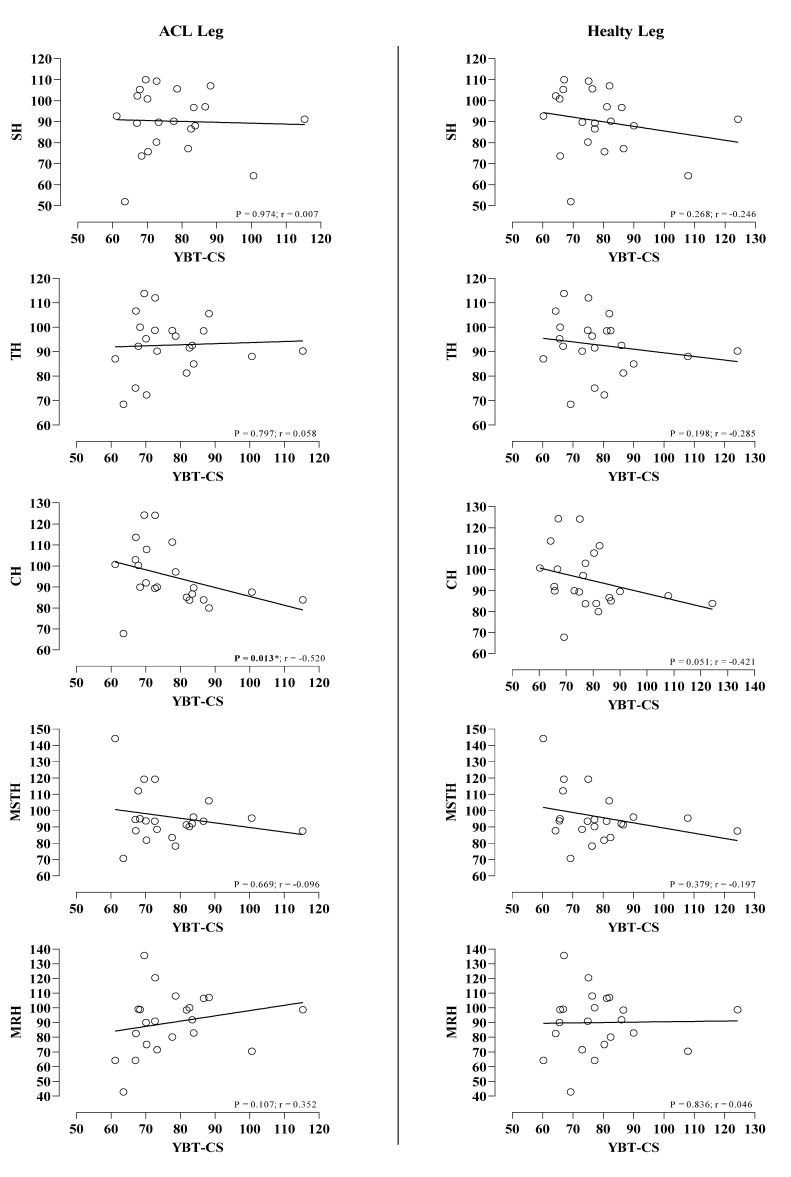
Correlation between YBT (composite score) and SLHT scores in operated (ACL-Leg) and healthy legs.

**Table 1 jpm-13-00466-t001:** Characteristic and clinical variables of the study cohort (n = 22).

Variables	Value
Age (years)	28.32 ± 6.6
Height (cm)	176.45 ± 6.68
Weight (kg)	86 ± 12.09
BMI (kg/m^2^)	27.55 ± 2.75
Time from ACLR to measurements (mon)	6.45 ± 0.67
Thigh length (cm)	
Operated side	47.27 ± 9.97
Non-operated side	47.58 ± 10.07
Lower limb length (cm)	
Operated side	90.5 ± 9.59
Non-operated side	90.73 ± 9.61
Dominant side (n (%))	
Right	17 (77.3)
Left	5 (22.7)
Surgical side (n (%))	
Right	13 (59.1)
Left	9 (40.9)

Data are presented as mean ± SD or *n* (%). ACLR, anterior cruciate ligament reconstruction; BMI, body mass index.

**Table 2 jpm-13-00466-t002:** Pre- and post-operative functional knee outcome scores (Lysholm, IKDC and Tegner).

	Pre-Operative		Post-Operative		95% CI	*p*-Value
	Mean ± SD	Med (Min-Max)	Mean ± SD	Med (Min-Max)		
Lysholm	74.09 ± 8.67	74 (58–92)	98.82 ± 2.56	100 (90–100)	–28.40 to−21.05	<0.001 ^1^
IKDC	49.27 ± 8.88	48.5 (32–64)	90.73 ± 6.32	91.5 (80–100)	–45.72 to−37.18	<0.001 ^2^
Tegner	6.36 ± 1.18	6 (5–9)	5.91 ± 1.31	6 (4–8)	0.229 to 0.681	0.002 ^1^

^1^ Wilcoxon test; ^2^ paired samples *t* test; IKDC, International Knee Documentation Committee; SD, standard deviation; Min, minimum; Max, maximum.

**Table 3 jpm-13-00466-t003:** YBT reach distances of the operated (ACL-Leg) and healthy leg (ANT, PL, and PM).

	ACL Leg	Healthy Leg	*p*-Value
	Median (Min-Max)	Median (Min-Max)	
Anterior (%)	72.54 (61.81–104.59)	74.68 (64.84–122.41)	0.123
Posterolateral (%)	67.98 (50.54–112.07)	74.57 (48.94–122.98)	0.592
Posteromedial (%)	79.88 (62.72–129.31)	82.16 (60.69–127.59)	0.884
Composite score (%)	72.96 (61.17–115.32)	76.76 (60.18–124.33)	0.291

Compares anterior, posterolateral, posteromedial, and composite scores for the operated and healthy legs. Although the all-direction reach scores of the operated leg were lower than the healthy leg, there were no statistically significant differences (*p* > 0.05). Min, minimum; max, maximum.

**Table 4 jpm-13-00466-t004:** Differences in the limb symmetry index (LSI).

Variables (%)	Mean ± SD	F	*p*-Value
SH	90.25 ± 15.06	0.677	0.610
TH	92.73 ± 11.79		
CH	95.1 ± 14.3		
MSTH	96.11 ± 15.98		
MRH	89.95 ± 20.69		

SD, standard deviation. SH, single hop for distance; TH triple hop for distance; CH crossover triple hop for distance; MSTH, medial side triple hop for distance; MRH medial rotation (90°) hop for distance.

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
