# Peer review of "The Relationship between Lower Extremity Functional Performance and Balance after Anterior Cruciate Ligament Reconstruction: Results of Patients Treated with the Modified All-Inside Technique"

_jpm, 2023, doi:10.3390/jpm13030466_

Round 1
Reviewer 1 Report
In the manuscript titled,Nizamettin Gü zel, Ahmet Serhat Genç, Ali Kerim Yılmaz demonstrated a retrospective cohort of 22 recreational athletes who underwent MAI ACLR techniques. This valuable study found that there was no significant difference in the function of the diseased limb between the diseased limb and the healthy limb 6 months after MAI ACL reconstruction. However, lack of innovation is the major flaw of the study. Therefore, MAJOR revision has to be done before this manuscript could be accepted for publication in the JPM.
Major comments :
1. The current manuscript needs to be polished by a native English speaker or a professional language editing service.
2. Please add the indications, gender characteristics of the included population.
3. Please describe the superiority of this surgical approach compared to other surgical approaches in the Discussion.
4. Please add intraoperative pictures and imaging data.
5. If possible, please give the study data of the contralateral limb before operation.
Minor comments :
1. Please add figure notes in figure1.
2. Change the test p-value to lowercase italics.
3. Please standardize the scientific notation in the table.
4. Please use English for lines 180-181 in the article.
Author Response
In the manuscript titled,Nizamettin Güzel, Ahmet Serhat Genç, Ali Kerim Yılmaz demonstrated a retrospective cohort of 22 recreational athletes who underwent MAI ACLR techniques. This valuable study found that there was no significant difference in the function of the diseased limb between the diseased limb and the healthy limb 6 months after MAI ACL reconstruction. However, lack of innovation is the major flaw of the study. Therefore, MAJOR revision has to be done before this manuscript could be accepted for publication in the JPM.
Reponse: Thank you very much for your comments and contributions to our research. We have tried to make all the corrections you have mentioned in our research. While we may not be able to fully respond to some of the corrections you requested, try to make all corrections as best we can. We are grateful for your time and contribution to our research.
Major comments:
- The current manuscript needs to be polished by a native English speaker or a professional language editing service.
Response: We had the English editing of the research done by an internationally certified firm. However, after the revision, as you said, we will review the whole revision from the beginning. Thanks a lot for your suggestion.
- Please add the indications, gender characteristics of the included population.
Response: As you suggested, we have stated the complaints of the patients when applying for the examination as indications. All of the patients were male, and we wrote this in the metho section. Data from all of the patients were presented, except for these, we did not receive any additional information from the patients. Thank you for your contributions.
- Please describe the superiority of this surgical approach compared to other surgical approaches in the Discussion.
Response: As you said, we have added the advantages of the modified all-inside technique to the discussion section. We are especially grateful to you for this warning, it was important to reveal the advantages of the technique we have applied.
- Please add intraoperative pictures and imaging data.
Response: Thank you for this unique contribution. We included the intraoperative photographs and postoperative radiological results available to the study. Thank you very much.
- If possible, please give the study data of the contralateral limb before operation.
Response: Thank you very much for your suggestions. Data on the contralateral limb are not available. However, we are aware of the importance of your suggestion and we will focus on this issue sensitively in future research. We are sorry that we could not answer your recommendation, thank you very much.
Minor comments:
- Please add figure notes in figure1.
Response: Added notes under figure for Figure 1. Thank you for your contribution.
- Change the test p-value to lowercase italics.
Response: All p values were changed as a italic. Thank you for your suggestion.
- Please standardize the scientific notation in the table.
Response: The tables were standardized according to the journal rules as you said.
- Please use English for lines 180-181 in the article.
Response: We are very sorry for our mistake. We are also grateful to you for your precise control. We revised it in English
Reviewer 2 Report
The paper by Nizamettin Güzel et al. addresses a very interesting subject. The authors report the results of a study regarding the relationship between lower extremity functional performance and balance after ACLR using a modified all inside technique. I would like to express sincere gratitude to get the opportunity to review your manuscript. The effort of the author is appreciated. The manuscript is well written, and the study protocols and data scientifically sound and are presented clearly. Congratulations on your results. There is a need to have large multicenter studies and hope to see data in the future. Maybe it would be interesting to evaluate the data also 1 year after surgery. Thank you for the performed literature review, no recommendations on this matter. I am curious were there any differences between the evaluated parameter in relation to the period until the surgery?
Author Response
The paper by Nizamettin Güzel et al. addresses a very interesting subject. The authors report the results of a study regarding the relationship between lower extremity functional performance and balance after ACLR using a modified all inside technique. I would like to express sincere gratitude to get the opportunity to review your manuscript. The effort of the author is appreciated. The manuscript is well written, and the study protocols and data scientifically sound and are presented clearly. Congratulations on your results. There is a need to have large multicenter studies and hope to see data in the future. Maybe it would be interesting to evaluate the data also 1 year after surgery. Thank you for the performed literature review, no recommendations on this matter. I am curious were there any differences between the evaluated parameter in relation to the period until the surgery?
Response: Thank you very much for your valuable comments on our research. We could not add what you said to the revision, as we did not receive any parameters from the patients in the time elapsed compared to the operation. However, if there is a different situation you want to describe, we are ready to review this situation and correct it again.
We are grateful for your contribution.
Best Regards.